# Changes in the Abundance of Danish Orchids over the Past 30 Years

**Christian Damgaard \*****, Jesper Erenskjold Moeslund and Peter Wind**

Department of Bioscience, Aarhus University, 8600 Silkeborg, Denmark; jesper.moeslund@bios.au.dk (J.E.M.); pwi@bios.au.dk (P.W.)

**\*** Correspondence: cfd@bios.au.dk

**Abstract:** Orchid abundance data collected over the past 30 years (1987–2016) from 440 sites within the National Orchid Monitoring Program were analyzed to quantify the population trends of orchids in Denmark, and the underlying reasons for the observed population trends were analyzed and discussed. Of the 45 monitored Danish orchids, 20 showed a significant decrease in abundance over the past 30 years (16, if only orchids with at least 50 observations each were selected), thus corroborating the previous observations of declining orchid abundances at the European scale. Generally, there was a significant negative effect of overgrowing with tall-growing herbs and shrubs on the abundance of Danish orchids, mainly caused by change of farming practices, as extensive management, such as grazing or mowing of light-open grassland areas, has decreased.

**Keywords:** Danish orchid species; Danish Red List; National Orchid Monitoring Program; Danish Orchid Database; citizen science; orchid monitoring; vegetation; population size; population trend; plant abundance; plant diversity; pressures on orchid sites; farming practice

---

## 1. Introduction

Orchids are generally in decline globally [1–3]. In Denmark, 33 taxa out of 51 species, subspecies, and varieties of orchids (see Table 1) are assessed as either extinct, threatened, or near threatened on the Danish Red List [4]. In the present study, we used orchid abundance data collected in the 30 years between 1987 and 2016 from 440 sites within the National Orchid Monitoring Program to quantify the population trends of orchids in Denmark. Additionally, we analyzed and discussed underlying reasons for the observed population trends.

**Table 1.** The 51 orchids recorded in Denmark, their taxonomic status [5], Danish Red List status [4], and period of monitoring of 45 orchids in the National Orchid Monitoring Program. Red List categories: RE = regionally extinct; CR = critically endangered; EN = endangered; VU = vulnerable; NT = near threatened; LC = least concern; NE = not evaluated.

| International Name | Taxon Status | Status 2020 | Monitoring Period |
|---|---|---|---|
| *Anacamptis morio* (L.) R.M. Bateman, Pridgeon & M.W. Chase | Species | NT | 1987–2016 |
| *Anacamptis pyramidalis* (L.) Rich. | Species | VU | 1987–2016 |
| *Cephalanthera damasonium* (Mill.) Druce | Species | VU | 1987–2016 |
| *Cephalanthera longifolia* (L.) Fritsch | Species | EN | 1987–2016 |

**Table 1.** *Cont.*

| International Name | Taxon Status | Status 2020 | Monitoring Period |
|---|---|---|---|
| *Cephalanthera rubra* (L.) Rich. | Species | CR | 1987–2016 |
| *Coeloglossum viride* (L.) Hartm. | Species | RE | |
| *Corallorhiza trifida* Châtel | Species | VU | 1987–2016 |
| *Cypripedium calceolus* L. | Species | VU | 1987–2016 |
| *Dactylorhiza incarnata* (L.) Soó subsp. *incarnata* | Subspecies | LC | 1987–2016 |
| *Dactylorhiza incarnata* (L.) Soo subsp. *lobelii* (Verm.) H.A. Pedersen | Subspecies | VU | |
| *Dactylorhiza incarnata* (L.) Soó subsp. *incarnata* var. *ochroleuca* (Boll) Hyl. | Variety | EN | 1987–2016 |
| *Dactylorhiza maculata* (L.) Soó subsp. *fuchsii* (Druce) Hyl. | Subspecies | LC | 1987–2016 |
| *Dactylorhiza maculata* (L.) Soó subsp. *maculata* | Subspecies | LC | 1987–2016 |
| *Dactylorhiza majalis* (Rchb.) P.F. Hunt & Summerh. subsp. *calcifugiens* H.A. Pedersen | Subspecies | NE | |
| *Dactylorhiza majalis* (Rchb.) P.F. Hunt & Summerh. subsp. *integrata* (E.G. Camus) H.A. Pedersen var. *integrata* | Variety | LC | 1987–2001, 2014–2016 |
| *Dactylorhiza majalis* (Rchb.) P.F. Hunt & Summerh. subsp. *integrata* (E.G. Camus) H.A. Pedersen var. *junialis* (Verm.) H.A. Pedersen | Variety | VU | 1997–2000, 2007–2008 |
| *Dactylorhiza majalis* (Rchb.) P.F. Hunt & Summerh. subsp. *lapponica* (Hartm.) H. Sund. | Subspecies | RE | |
| *Dactylorhiza majalis* (Rchb.) P.F. Hunt & Summerh. subsp. *majalis* | Subspecies | LC | 1987–2016 |
| *Dactylorhiza majalis* (Rchb.) P.F. Hunt & Summerh. subsp. *purpurella* (T. & T.A. Stephenson) D.M. Moore & Soó var. *cambrensis* (R.H. Roberts) H.A. Pedersen | Variety | LC | 1987–2003 |
| *Dactylorhiza majalis* (Rchb.) P.F. Hunt & Summerh. subsp. *purpurella* (T. & T.A. Stephenson) D.M. Moore & Soó var. *purpurella* | Variety | LC | 1987–2016 |
| *Dactylorhiza majalis* (Rchb.) P.F. Hunt & Summerh. subsp. *sphagnicola* (Höppner) H.A. Pedersen & Hedrén | Subspecies | CR | |
| *Dactylorhiza sambucina* (L.) Soó | Species | EN | 1987–2016 |
| *Epipactis atrorubens* (Hoffm. ex Bernh.) Besser | Species | NT | 1989–2010 |
| *Epipactis helleborine* (L.) Crantz subsp. *helleborine* | Subspecies | LC | 1987–2016 |
| *Epipactis helleborine* (L.) Crantz subsp. *neerlandica* (Verm.) Buttler var. *neerlandica* | Variety | LC | 1990–1993, 2007–2016 |
| *Epipactis helleborine* (L.) Crantz subsp. *neerlandica* (Verm.) Buttler var. *renzii* (Rob.) J.Claess., Kleynen & Wielinga | Variety | NT | |
| *Epipactis leptochila* (Godfery) Godfery | Species | NT | 1987–2012 |
| *Epipactis palustris* (L.) Crantz | Species | NT | 1987–2016 |
| *Epipactis phyllanthes* G.E. Sm. | Species | LC | 1989–2016 |
| *Epipactis purpurata* Sm. | Species | LC | 1987–2016 |
| *Epipogium aphyllum* Sw. | Species | CR | 1987–2012 |
| *Goodyera repens* (L.) R. Br. | Species | LC | 1987–2000, 2010–2016 |
| *Gymnadenia conopsea* (L.) R. Br. subsp. *conopsea* | Subspecies | CR | 1987–2016 |
| *Gymnadenia conopsea* (L.) R. Br. subsp. *densiflora* (Wahlenb.) K Richt. | Subspecies | CR | 1998–2011, 2015–2016 |
| *Hammarbya paludosa* (L.) Kuntze | Species | EN | 1987–2016 |
| *Herminium monorchis* (L.) R. Br. | Species | EN | 1987–2016 |
| *Liparis loeselii* (L.) Rich. | Species | EN | 1987–2016 |
| *Neotinea ustulata* (L.) R.M. Bateman, Pridgeon & M.W. Chase | Species | CR | 1987–1991, 1994–2003, 2008–2016 |

**Table 1.** *Cont.*

| International Name | Taxon Status | Status 2020 | Monitoring Period |
|---|---|---|---|
| *Neottia cordata* (L.) Rich. | Species | LC | 1989–2011 |
| *Neottia nidus–avis* (L.) Rich. | Species | LC | 1987–2016 |
| *Neottia ovata* (L.) Bluff & Fingerh. | Species | LC | 1987–2016 |
| *Ophrys apifera* Huds. | Species | VU | 2004–2016 |
| *Ophrys insectifera* L. | Species | CR | 1987–1995, 2005–2009 |
| *Orchis mascula* (L.) L. | Species | LC | 1987–2016 |
| *Orchis militaris* L. | Species | NA | 1987–1994 |
| *Orchis purpurea* Huds. | Species | NT | 1987–2016 |
| *Platanthera bifolia* (L.) Rich. subsp. *bifolia* | Subspecies | NT | 1987–2016 |
| *Platanthera bifolia* (L.) Rich. subsp. *latiflora* (Drejer) Løjtnant | Subspecies | EN | 1987–2016 |
| *Platanthera chlorantha* (Custer) Rchb. | Species | NT | 1987–2016 |
| *Pseudorchis albida* (L.) À & D. Löve | Species | CR | 1987–2016 |
| *Spiranthes spiralis* (L.) Chevall. | Species | RE | 1987–1991, 1996–1999 |

In Northern Europe, orchids typically grow in woodlands, both broad-leaved forests and old conifer plantations, dry grasslands and heathlands, rich and poor fens, meadows, and dune slacks [5]. Sometimes, orchids also colonize abandoned marl and lime pits, but avoid saline marshes, arid white dune areas and intensively managed farmlands [5]. The soil's structure and texture, hydrology, pH, mycorrhiza availability, and composition of nutrients are important ecological factors that characterize suitable habitats for orchids [6–9]. Furthermore, light conditions, the degree of disturbance, and, for some orchids, the presence of specific pollinators play important roles for habitat suitability [5,10].

Most orchids in Northern Europe have declined because of habitat loss, habitat fragmentation, or habitat degradation, for example as a result of secondary succession following abandonment of former farming practices, such as extensive grazing with livestock and mowing [1,11–13]. The secondary succession can accelerate by changed interspecific competitive regimes following increased soil nutrient availability due to fertilizers and airborne nitrogen pollution [14]. Other land use changes, such as urbanization, infrastructural constructions, lowering of the groundwater table, draining, and cultivation, are equally important causes of orchid habitat loss [1,12]. In some cases, overgrazing can also constitute a threat to orchids [15], e.g., by preventing seed development. In forests, many orchids grow in areas that have been left untouched for decades, and if the forests are disturbed, e.g., drained, clear-cut, or fertilized, then orchids are at risk of disappearing [16]. Despite being protected by law in many countries, including Denmark [17], wherein digging or pricking of specimens, collecting seeds, or in any way harming the individual specimen is prohibited, orchids are regularly picked for flower bouquets or dug up for gardening or trade, actions that both affect the fitness of an orchid population. For instance, three entire clones of *Cypripedium calceolus* were dug up from the Danish population at Buderupholm in 2012 and moved from a fence that had been raised in order to protect the population, and in 2016, approximately 40 flowering shoots were picked from the population (Himmerland Forest Department, pers.com. 2012, 2016). Additionally, general public disturbance and outdoor activities, such as hiking, camping, and picnicking, may also affect orchid populations negatively [18–23]. For example, an *Epipactis leptochila* population at a slope on Møns Klint possibly went extinct due to the playing of children (N. Faurholdt, per.com 1996). Many orchid species are specialists, depending on only one or a few pollinators or mycorrhizal symbiont species [24,25]. This can explain why orchids are intrinsically rare [26], but has probably also given orchids their reputation for being good indicators of intact or high-value nature [27,28].

The monitoring of orchid species provides a good basis for effective management and conservation of native orchid populations and their habitats, which is a complicated task due to the orchid species' complex ecology and the many and varied threats mentioned above. The annual censuses of orchid populations performed within the National Orchid Monitoring Program is an example of such monitoring that provides important information on abundance trends of the Danish orchid species.

Moreover, the program is an excellent example of an ongoing citizen science project, where Danish orchids have been monitored annually for more than 30 years from 440 selected orchid sites. The National Orchid Monitoring program is, to our knowledge, the most comprehensive and long-lived field-based orchid monitoring project in the world comprising most Danish members of the orchid family (see Supplementary Materials Appendix SA for a detailed account of citizen science in Denmark).

Internationally, there are a few examples of long-term orchid monitoring programs of comparable duration as the monitoring initiated in 1943 of the Danish population of *Cypripedium calceolus*. One example is the annual census of a population of *Anacamptis pyramidalis* in a Dutch dune area that started in 1940 [29]. The annual census was conducted continuously until at least 1975, except for the World War years of 1943, 1944, and 1945 [29]. Another example is the Swedish ecologist C.O. Tamm (1919–2007), who laid out the first plot in 1942 in a population of *Dactylorhiza sambucina* in order to document the effect of picking orchids [30]. The annual Swedish census also involved other orchid species—*Dactylorhiza incarnata*, *Neottia ovata*, and *Orchis mascula*. The monitoring seems to have ended in 1990. Additionally, long-time monitoring has been performed on *Orchis anthropophora* (L.) All. 1967–1980 [31], *Dactylorhiza majalis* subsp. *integrata* 1973–1985 (syn. *D. praetemissa*), [32,33], *Dactylorhiza sambucina* 1968–1985 [12], *Orchis mascula* 1960–1970 [34], *O. militaris* 1947–1962 [35,36], *O. palustris* [37], and *Spiranthes spiralis* [31,38]. A study of 47 orchid species at 26 sites with three different habitat types—grassland, shrubland and woodland—was performed between March and May for an 8-year period (2006–2013) in the Mediterranean part of France, where the flowering plants were recorded on sites ranging between 500 and 2000 m$^2$ in size [39].

Kull [40] made an overview of long-term field-based studies on population dynamics of terrestrial orchids. The overview comprises 66 orchids with information on the ways of performing the monitoring: (1) monitoring in permanent plots; (2) counting of specimens in populations; (3) genet dynamics, where individuals are mapped; (4) measurement of fruit set; and (5) morphometrical parameters analyzed. A characteristic feature of these studies is that they are generally short-termed compared to the National Orchid Monitoring Program and most studies only monitor a single or a few species.

There is a need for monitoring data to support decision making in nature conservation and restoration, especially for species groups that are good indicators of the status of nature. In this study, we demonstrated the importance of long-term monitoring of such species for nature management. More specifically, we used data from the long-term (30 years) National Orchid Monitoring Program to address the following questions: Which orchid species are declining and which are not? What is the degree of decline? What are the most important pressures that can explain the population dynamics that we have observed?

## 2. Materials and Methods

### 2.1. Orchid Taxa in Denmark

A total of 51 terrestrial orchids have been recorded in Denmark [5] and belong to 10 boreal genera of the orchid family (*Orchidaceae*). All of them are native to the Danish flora and comprise 31 species, 13 subspecies, and 7 varieties. Two species, *Orchis militaris* with 1 flowering plant in a calcareous pit and *Ophrys apifera* with 50 flowering plants in an abandoned marl pit, were recorded for the first time in Denmark in 1981 and 2004, respectively [41,42]. In 1989, the population of *O. militaris* was destroyed as a result of replenishment and planning work in part of the lime pit [5]. The orchid was rediscovered with one flowering plant in another part of Denmark in 2016 [43].

In the following, the term "orchid/orchids" refers to taxa at the species level and lower levels, unless otherwise specified, while the nomenclature follows Pedersen and Faurholdt [5].

### 2.2. Status, Biology, and Habitats of Danish Orchids

Undoubtedly, the Danish orchids belong to one of the most threatened group of vascular plants in the Danish flora. Out of the 51 (65%) Danish orchids assessed for the Danish Red List [4], 33 are either

extinct, threatened, or near threatened by extinction on the basis of the IUCN Red List Assessment methodology. In addition, orchids constitute 10% of the 324 threatened Danish taxa. A total of 8 orchids are categorized as critically endangered (CR), 7 endangered (EN), 7 vulnerable (VU), and 8 near threatened (NT). A total of 16 orchids are categorized least concern (LC), while 2 have not yet been assessed. Moreover, 2 species and 1 subspecies are now regionally extinct (RE) in Denmark (Table 1) [4].

The rareness and threatened status are also reflected in the national mapping of the vascular plants in Denmark, *Atlas Flora Danica* [44], where the extant Danish orchids have been recorded from *Epipogium aphyllum* in 1 to *Epipactis helleborine* subsp. *helleborine* in 677 grid-squares of 5 × 5 km$^2$. The number of orchid sites varies from one (various orchids) to *Dactylorhiza majalis* subsp. *Majalis*, which are present on at least 830 sites (Table 2).

**Table 2.** Status, biology, and habitats of Danish orchids. The number in parenthesis in "no. grid-squares" is the number cited in Hartvig [44], while the next number without parenthesis is the actual number. The estimated number of sites is based on unpublished data provided by P. Wind. Information on biology and habitats is based on Pedersen and Faurholdt [5] and Hartvig [44]. Abbreviations of orchid names: *Ana mor—Anacamptis morio, Ana pyr—A. pyramidalis, Cep dam—Cephalanthera damasonium, Cep lon—C. longifolium, Cep rub—C. rubra, Coe vir—Coeloglossum viride, Cor tri—Corallorhiza trifida, Cyp cal—Cypripedium calceolus, Dac inc inc—Dactylorhiza incarnata* subsp. *incarnata, Dac inc inc och—D. i.* subsp. *i.* var. *ochroleuca, Dac mac fuc—D. maculata* subsp. *fuchsii, Dac mac mac—D. m.* subsp. *maculata, Dac maj int int—D. majalis* subsp. *integrata* var. *integrata, Dac maj int jun—D. m.* subsp. *i.* var. *junialis, Dac maj lap—D. m.* subsp. *lapponica, Dac maj maj—D. m.* subsp. *majalis, Dac maj pur cam—D. m.* subsp. *purpurella* var. *cambrensis, Dac maj pur pur—D. m.* subsp. *p.* var. *purpurella, Dac sam—D. sambucina, Epi atr—Epipactis atrorubens, Epi hel hel—E helleborine* subsp. *helleborine, Epi hel nee—E. h.* subsp. *neerlandica, Epi lep—E. leptochila, Epi pal—E. palustris, Epi phy—E. phyllanthes, Epi pur—E. purpurata, Epi aph—Epipogium aphyllum, Goo rep—Goodyera repens, Gym con con—Gymnadenia conopsea* subsp. *conopsea, Gym con den—G. c.* subsp. *densiflora, Ham pal—Hammarbya paludosa, Her mon—Herminium monorchis, Lip loe—Liparis loeselii, Neoti ust—Neotinea ustulata, Neo cor—Neottia cordata, Neo nid—N. nidus-avis, Neo ova—N. ovata, Oph api—Ophrys apifera, Oph ins—O. insectifera, Orc mas—Orchis mascula. Orc mil—O. militaris, Orc pur—Orchis purpurea, Pla bif bif—Platanthera bifolia* subsp. *latiflora, Pla bif lat—P. b.* subsp. *latiflora, Pla chl—P. chlorantha, Pse alb—Pseudorchis albida. Spi spi—Spiranthes spiralis.*

| Orchid Name | No. Grid-Squares | No. of Sites | Underground Part | Vegetative Propagation | Fertilisation | Production of Nectar | Pimary Habitat(s) |
|---|---|---|---|---|---|---|---|
| *Ana mor* | 26 | 34 | Tuber | Formation of more tubers | Cross-fertilization | No | Grassland |
| *Ana pyr* | (2) 3 | 4 | Tuber | Formation of more tubers | Cross-fertilization | No | Calcareous grassland |
| *Cep dam* | 20 | 21 | Rhizome | Division of rhizome | Self-fertilization | No | Deciduous woodland on chalk |
| *Cep lon* | 13 | 13 | Rhizome | Division of rhizome | Cross-fertilization | No | Deciduous woodland on chalk |
| *Cep rub* | 3 | 5 | Rhizome | Division of rhizome | Cross-fertilization | No | Deciduous woodland on chalk |
| *Coe vir* | 0 | 0 | Tuber | Formation of more tubers | Cross-fertilization | Yes | Grassland |
| *Cor tri* | 20 | 21 | Rhizome | Division of rhizome | Cross- and self-fertilization | No | Deciduous woodland on chalk, wooded poor fen |
| *Cyp cal* | 2 | 2 | Rhizome | Division of rhizome | Cross-fertilization | No | Deciduous woodland on chalk, calcareous grassland |

**Table 2.** *Cont.*

| Orchid Name | No. Grid-Squares | No. of Sites | Underground Part | Vegetative Propagation | Fertilisation | Production of Nectar | Pimary Habitat(s) |
|---|---|---|---|---|---|---|---|
| *Dac inc inc* | 338 | 384 | Tuber | Formation of more tubers | Cross-fertilization | No | Rich fen |
| *Dac inc lob* | 4 | 7 | Tuber | Formation of more tubers | Cross-fertilization | No | Rich fen |
| *Dac inc inc och* | (4) 3 | 3 | Tuber | Formation of more tubers | Cross-fertilization | No | Rich fen |
| *Dac mac fuc* | 71 | 130 | Tuber | Formation of more tubers | Cross-fertilization | No | Deciduous woodland |
| *Dac mac mac* | 268 | 424 | Tuber | Formation of more tubers | Cross-fertilization | No | Grassland, heathland, fen |
| *Dac maj cal* | 7 | 10 | Tuber | Formation of more tubers | Cross-fertilization | No | Dune slack, rich fen |
| *Dac maj int int* | 9 | 11 | Tuber | Formation of more tubers | Cross-fertilization | No | Rich fen |
| *Dac maj int jun* | 4 | 7 | Tuber | Formation of more tubers | Cross-fertilization | No | Rich fen |
| *Dac maj lap* | 0 | 0 | Tuber | Formation of more tubers | Cross-fertilization | No | Fen |
| *Dac maj maj* | 556 | 830 | Tuber | Formation of more tubers | Cross-fertilization | No | Rich fen |
| *Dac maj pur cam* | 12 | 30 | Tuber | Formation of more tubers | Cross-fertilization | No | Dune slack, rich fen |
| *Dac ma pur pur* | 73 | 60 | Tuber | Formation of more tubers | Cross-fertilization | No | Dune slack, rich fen |
| *Dac maj sph* | 3 | 3 | Tuber | Formation of more tubers | Cross-fertilization | No | Poor fen |
| *Dac sam* | 7 | 13 | Tuber | Formation of more tubers | Cross-fertilization | No | Grassland, heathland |
| *Epi atr* | 2 | 3 | Rhizome | Division of rhizome | Cross-fertilization | Yes | Conifer and deciduous woodland on chalk, grassland |
| *Epi hel hel* | 677 | 701 | Rhizome | Division of rhizome | Cross-fertilization | Yes | Deciduous woodland, grassland |
| *Epi hel nee nee* | 19 | 30 | Rhizome | Division of rhizome | Cross-fertilization | Yes | Calcareous dune |
| *Epi hel nee ren* | 3 | 4 | Rhizome | Division of rhizome | Self-fertilization | No | Calcareous dune |
| *Epi lep* | 7 | 11 | Rhizome | Division of rhizome | Self-fertilization | No | Deciduous woodland |
| *Epi pal* | 123 | 231 | Rhizome | Division of rhizome | Cross-fertilization | Yes | Dune slack, rich fen |
| *Epi phy* | 70 | 119 | Rhizome | Division of rhizome | Self-fertilization | No | Deciduous woodland |
| *Epi pur* | 23 | 52 | Rhizome | Division of rhizome | Cross-fertilization | Yes | Deciduous woodland |
| *Epi aph* | 1 | 1 | Rhizome | Division of rhizome | Cross-fertilization | Yes | Deciduous woodland |
| *Goo rep* | 17 | 18 | Rhizome | Division of rhizome | Cross-fertilization | Yes | Conifer forest |

**Table 2.** *Cont.*

| Orchid Name | No. Grid-Squares | No. of Sites | Underground Part | Vegetative Propagation | Fertilisation | Production of Nectar | Pimary Habitat(s) |
|---|---|---|---|---|---|---|---|
| *Gym con con* | (4) 1 | 1 | Tuber | Formation of more tubers | Cross-fertilization | Yes | Rich fen |
| *Gym con den* | (1) 2 | 2 | Tuber | Formation of more tubers | Cross-fertilization | Yes | Rich fen |
| *Ham pal* | 18 | 21 | Pseudobulb | Formation of more pseudobulbs and bulbils at leaftips | Cross-fertilization | Yes | Poor fen, blanket bog |
| *Her mon* | 10 | 18 | Tuber | Formation of more tubers | Cross-fertilization | No | Dune slack, rich fen |
| *Lip loe* | 15 | 20 | Pseudobulb | Formation of more pseudobulbs | Cross- and self-fertilization | No | Dune slack, rich fen |
| *Neoti ust* | 2 | 3 | Tuber | Formation of more tubers | Cross-fertilization | No | Calcareous grassland |
| *Neo cor* | 18 | 21 | Rhizome | Division of rhizome | Cross-fertilization | Yes | Conifer forest, wooded poor fen |
| *Neo nid* | 119 | 149 | Rhizome | Division of rhizome | Cross- and self-fertilization | No | Deciduous woodland |
| *Neo ova* | 306 | 495 | Rhizome | Division of rhizome | Cross-fertilization | Yes | Deciduous woodland, grassland, rich fen |
| *Oph api* | 2 | 5 | Tuber | Formation of more tubers | Self-fertilization | No | Grassland |
| *Oph ins* | 2 | 1 | Tuber | Formation of more tubers | Cross-fertilization | No | Calcareous grassland |
| *Orc mas* | 301 | 576 | Tuber | Formation of more tubers | Cross-fertilization | No | Deciduous woodland, grassland |
| *Orc mil* | (0) 1 | 1 | Tuber | Formation of more tubers | Cross-fertilization | No | Calcareous grassland |
| *Orc pur* | 9 | 10 | Tuber | Formation of more tubers | Cross-fertilization | No | Deciduous woodland on chalk, calcareous grassland |
| *Pla bif bif* | 90 | 123 | Tuber | Formation of more tubers | Cross-fertilization | Yes | Grassland, heathland |
| *Pla bif lat* | 3 | 3 | Tuber | Formation of more tubers | Cross-fertilization | Yes | Deciduous woodland on chalk |
| *Pla chl* | 231 | 374 | Tuber | Formation of more tubers | Cross-fertilization | Yes | Deciduous woodland, grassland |
| *Pse alb* | (6) 3 | 3 | Tuber | Formation of more tubers | Cross- and self-fertilization | No | Grassland |
| *Spi spi* | 0 | 0 | Tuberous root | Division of tuberous roots | Cross-fertilization | Yes | Grassland |

All Danish orchids are terrestrial perennials supplied with either tubers (31 orchids), rhizomes (18 orchids), pseudobulbs (2 orchids), or tuberous roots by the extinct *Spiranthes spiralis*—all subterranean organs that are responsible for storing of nutrients for the following year's formation of over-ground parts such as aerial stems, leaves, and inflorescences. During the summer season, a new tuber or a pseudobulb substitutes the old one. In some instances, more tubers or pseudobulbs are formed,

giving rise to more new independent daughter offspring genetically identical with the mother plant the following year. The rhizome can branch and each branch is able to form aerial shoots that are impossible to separate from each other without digging the rhizome complex up and, thus, functions as a clone. All aerial shoots are therefore counted separately.

Most Danish orchids have conspicuous flowers, which makes it easy to identify and determine the particular orchids in a dense vegetation cover, especially in species-rich and species-dense fens. A few have more inconspicuous flowers, e.g., *Hammerbya paludosa* and *Pseudorchis albida*, and can easily be overlooked, especially the latter when growing together with *Platanthera* spp., a problem that Voigt-Schilb et al. [45] have also realized.

Pollination of the orchid flowers implies that the individual plants are able to produce capsules with ripe seeds. The fertilization is carried out by obligate cross-pollination (42 orchids), although more than half of the species produce no nectar that can attract pollinators. Five orchids are obligate self-pollinators. The flowers of four orchids are originally designed for cross-pollination, but due to various mechanisms in the flower, self-pollination takes over (Table 2), which the Danish botanist Hagerup [46] has demonstrated for *Liparis loeselii*.

Many Danish orchids have narrow ecologically requirements, as they are confined to only one habitat type as shade-requiring under more protected, shaded conditions in woodland; or as shade-intolerant under light-open, moist, or dry calcareous conditions. As pointed out by Swarts and Dixon [47], the adaptability to one or more mycorrhizae, the attractiveness of the orchid flowers for one or more pollinators, and the ability for dispersal of the ripe seeds are all parameters that limit the distribution and the different orchids' choice of habitat.

Even though the Danish terrestrial area at 43,000 km$^2$ is quite small compared to many other countries, there is a great difference in the annual precipitation between the different parts of Denmark. Precipitation is lowest in the coastal areas, especially in the Storebælt area, and highest in the interior parts, which is reflected in the distribution pattern of many plant species and orchids as well. Additionally, the composition of the soil influences the distribution of plants, including orchids. The western part of Jutland was ice-free during the latest Weichel Ice Age, resulting in a sandier, nutrient-poor soil, while clay or calcareous moraine cover most of the remaining areas of Denmark.

Concerning the choice of habitat, 14 Danish orchids grow solely in rich fens and dune slacks, while 4 of them are restricted to calcareous grassland and 2 to calcareous dunes. Thus, all 18 orchids can be characterized as shade-intolerant. Three orchids are found in deciduous woodland on chalk only, while 6 others are limited to other types deciduous woodland, especially beech forest, and thus can be characterized as shade-requiring. The rest of the Danish orchids have more broad ecological requirements and can be found in different habitat types as deciduous woodland and on grassland both with and without calcareous soil (Table 2).

*2.3. Orchid Censuses in Denmark*

The census of Danish orchid populations started long before the onset of the National Orchid Monitoring Program in 1987. The monitoring of orchids expanded in the beginning of the 1980s, where three Danish counties began to perform an annual census of local populations of orchids such as *Anacamptis morio*, *Dactylorhiza maculata* subsp. *fuchsii*, *Dactylorhiza majalis* subsp. *majalis*, *Epipactis palustris*, *Gymnadenia conopsea* subsp. conopsea, and *Liparis loeselii*. At the same time, amateurs started regular annual censuses of selected populations of some of the rarest Danish orchids, e.g., *Anacamptis pyramidalis*, *Epipactis leptochila*, *Epipogium aphyllum*, and *Pseudorchis albida* (Figure 1).

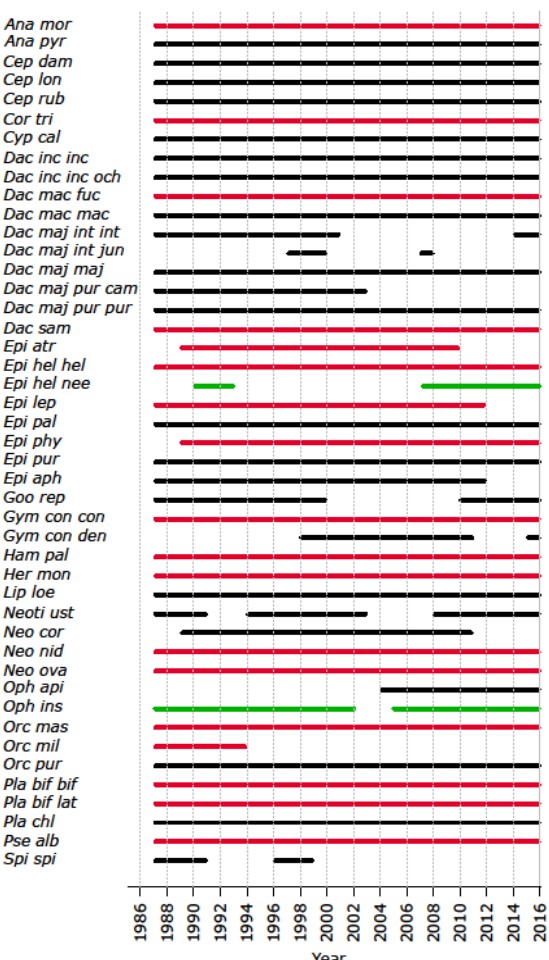

**Figure 1.** Onset and continuity of the annual census of 45 Danish orchids that is based on the data in the Danish Orchid Database. Bars in red indicate that the number of flowering shoots of the species in question is in decline, black means that it is stable, and green that it is increasing. Abbreviations: *Ana mor*—*Anacamptis morio*, *Ana pyr*—*A. pyramidalis*, *Cep dam*—*Cephalanthera damasonium*, *Cep lon*—*C. longifolium*, *Cep rub*—*C. rubra*, *Cor tri*—*Corallorhiza trifida*, *Cyp cal*—*Cypripedium calceolus*, *Dac inc inc*—*Dactylorhiza incarnata* subsp. *incarnata*, *Dac inc inc och*—*D. i.* subsp. *i.* var. *ochroleuca*, *Dac mac fuc*—*D. maculata* subsp. *fuchsii*, *Dac mac mac*—*D. m.* subsp. *maculata*, *Dac maj int int*—*D. majalis* subsp. *integrata* var. *integrata*, *Dac maj int jun*—*D. m.* subsp. *i.* var. *junialis*, *Dac maj maj*—*D. m.* subsp. *majalis*, *Dac maj pur cam*—*D. m.* subsp. *purpurella* var. *cambrensis*, *Dac maj pur pur*—*D. m.* subsp. *p.* var. *purpurella*, *Dac sam*—*D. sambucina*, *Epi atr*—*Epipactis atrorubens*, *Epi hel hel*—*E helleborine* subsp. *helleborine*, *Epi hel nee*—*E. h.* subsp. *neerlandica*, *Epi lep*—*E. leptochila*, *Epi pal*—*E. palustris*, *Epi phy*—*E. phyllanthes*, *Epi pur*—*E. purpurata*, *Epi aph*—*Epipogium aphyllum*, *Goo rep*—*Goodyera repens*, *Gym con con*—*Gymnadenia conopsea* subsp. *conopsea*, *Gym con den*—*G. c.* subsp. *densiflora*, *Ham pal*—*Hammarbya paludosa*, *Her mon*—*Herminium monorchis*, *Lip loe*—*Liparis loeselii*, *Neoti ust*—*Neotinea ustulata*, *Neo cor*—*Neottia cordata*, *Neo nid*—*N. nidus-avis*, *Neo ova*—*N. ovata*, *Oph api*—*Ophrys apifera*, *Oph ins*—*O. insectifera*, *Orc mas*—*Orchis mascula.* *Orc mil*—*O. militaris*, *Orc pur*—*Orchis purpurea*, *Pla bif bif*—*Platanthera bifolia* subsp. *latiflora*, *Pla bif lat*—*P. b.* subsp. *latiflora*, *Pla chl*—*P. chlorantha*, *Pse alb*—*Pseudorchis albida. Spi spi*—*Spiranthes spiralis*.

The annual census of 18 Danish orchids was performed at the start in 1987 of the National Orchid Monitoring Program, including the populations of orchids that had already been monitored, as mentioned above. The annual census was continued for 14 of the 18 orchids throughout the first 30 years of the program. In 1989 and in 1990, 4 additional orchids were included in the program, of which the annual censuses were performed for 1 of them, *Epipactis phyllanthes*, during the rest of

the period. For the 3 other orchids, the censuses of *Epipactis atrorubens* and *Neottia cordata* ended in 2009 and 2011, respectively, and those of *Epipactis helleborine* subsp. *neerlandica* were performed more irregularly on different sites (Figure 2). The annual censuses of *Ophrys apifera* started in 2004, the year the orchid species was observed for the first time in Denmark. Thus, 45 Danish orchids (30 species, 9 subspecies, and 6 varieties) have been monitored either continuously (31 orchids) or more irregularly. The latter was either because monitoring stopped (10 orchids) during the period 1987–2016 or since their first inclusion in the program. A full overview of the monitored orchids and their census periods can be found in Table 1 and Figure 2, respectively.

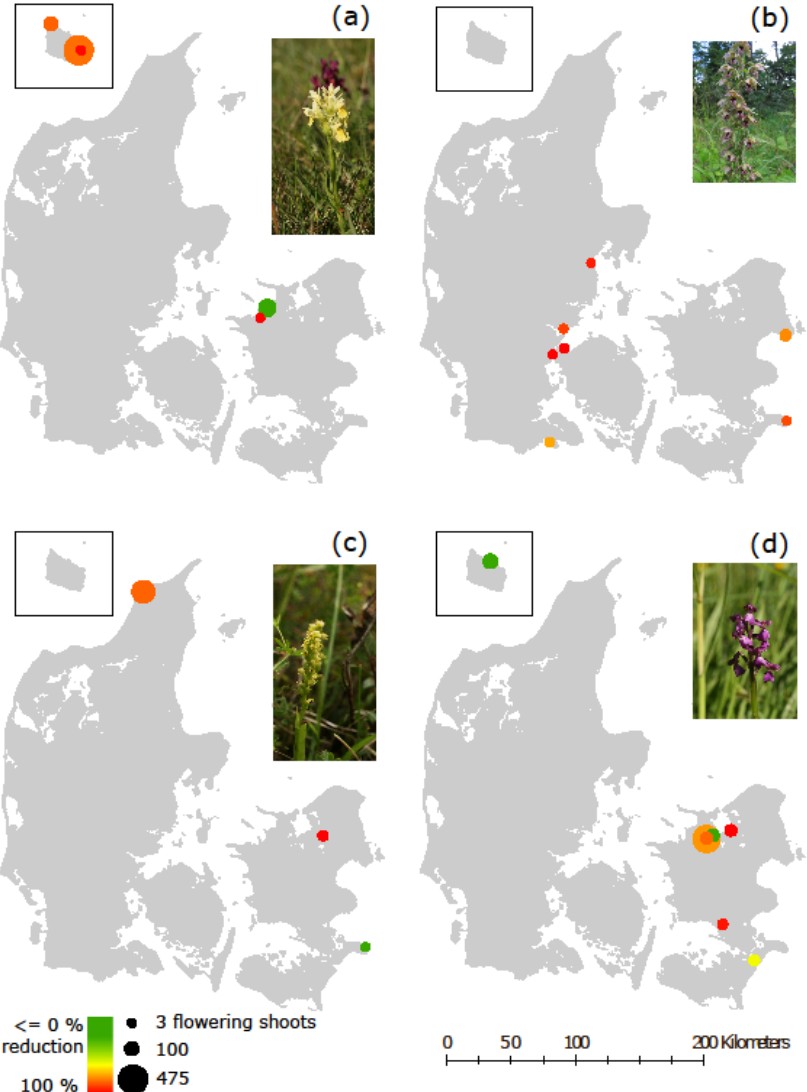

**Figure 2.** Populations of the four analyzed orchids. Only populations that were monitored at least 3 years within the first and last 10 years of the study period are shown. The size of the dots corresponds to the mean number of flowering shoots (1987–1996), showing the initial population sizes. The color gradient from green to red reflects the reduction of flowering shoots from the initial period (1987–1996) to the 2007–2016 period. The insert shows the island of Bornholm. (**a**) *Dactylorrhiza sambucina*, (**b**) *Epipactis helleborine* subsp. *helleborine*, (**c**) *Herminium monorchis*, (**d**) *Anacamptis morio*. Photos: (**a,c,d**) Jesper Moeslund. (**b**) Peter Wind.

Two Danish orchid species, *Cypripedium calceolus* and *Liparis loeselii*, are included in the European Union Habitats Directive Annex 2 [48], and all known Danish populations of the two species have been monitored annually since the onset of the national monitoring program for species, and terrestrial and aquatic nature in 2004.

## 2.4. Orchid Abundance Data

Local volunteers are of great importance in the National Orchid Monitoring Program, as they perform the annual census of selected orchid populations. The surveyors in the program also include paid field biologists. All participants possess in-depth knowledge of the orchids. Not all Danish orchid sites were included in the program; many sites were selected in close distance to the surveyors' residences or they were selected on the basis of their knowledge of the locations of the most important local populations. Only sites with more than 30 flowering orchid shoots at the start of the program were included [49]. On many sites, different orchids grow together, e.g., on fen sites comprising the 2 orchids *Dactylorhiza incarnata* subsp. *incarnata* and *D. majalis* subsp. *majalis*. In such instances, both species were monitored. At each of the selected 440 sites, the number of flowering shoots, either at the whole site or at a permanent plot within the site, were counted in the flowering season. A sub-sample of the collected abundance data is illustrated for four orchids (Figure 2).

The surveyors also estimated the intensity of grazing with livestock, forest management, overgrowing with tall-growing herbs and shrubs, and public disturbance at the orchid sites by using a four-step classification of the pressures, i.e., (1) none, (2) weak, (3) moderate, and (4) hard or strong [50].

Further details on the National Orchid Monitoring Program are compiled in Supplementary Materials, Appendix SB.

On 8 August 2019, the existing 9688 records (observations) of flowering shoots from 874 Danish orchid populations at the 440 orchid sites, as well as 3337 records on pressures, were retrieved from the Danish Orchid Database and provide the basis for the present analysis.

## 2.5. Statistical Analysis

The observed changes in orchid species abundances were modelled on the basis of the recorded number of flowering shoots, and since the data structure was irregular, it was decided to model the change in abundance by a state-space model, where the species abundance at a specific site $i$ at time $t$ was modelled by latent variables, $x_{i,t}$. The observed abundance is denoted by $y_{i,t}$ and is assumed to be Poisson distributed with the latent variables as the mean parameters, $y_{i,t} \sim Poisson(x_{i,t})$. The change in the log-transformed abundance was modelled by two linear models,

$$log(x_{i,t}) = \alpha + \vartheta_i + \beta t \tag{1}$$

$$log(x_{i,t}) = \alpha + \vartheta_i + (\beta + \delta_i)t \tag{2}$$

where $\alpha$ is the intercept and $\beta$ is the mean annual change. The random effects of site and site * years are modelled by $\vartheta_i$ and $\delta_i$, respectively, and both are assumed to be normally distributed. The two linear models differ in whether the random effect of site * years is included or not, i.e., whether the change in abundance varies among sites. Due to the log-transformation, the estimated doubling time of a population may be calculated as $log(2)/\beta$ or $log(2)/(\beta + \delta_i)$ in model 1 and 2, respectively.

The models were fitted in a Bayesian framework using integrated nested Laplace approximation (INLA) [51]. The implementation of the two models in R is shown in Supplementary Materials, Appendix SD and follows Blangiardo et al. [52]. The two models were compared by DIC [53], and statistical inferences were made using the 95% credible interval of the parameter of interest.

In order to investigate possible causal relationships between the abiotic environment and the observed site-specific changes in abundance, we regressed the estimated site-specific random time

coefficients, $\delta_i$, of model 2 a+gainst the mean values of the four estimated pressures at each site with species as a random effect. Again, this model was fitted using INLA [51].

## 3. Results

Generally, the two linear models (1 and 2) gave the same qualitative results, i.e., the estimated trends, $\beta$, had approximately the same credibility interval. However, model 2 better supported the abundance data for the majority of the orchid species (Table 3), i.e., for most species, the rate of change differed significantly among sites.

**Table 3.** The number of observations, N; percentiles of the marginal distribution of the average change over time (bold green numbers denote a significant increase and bold red numbers denote a significant decrease), $\beta$; and the model (Equation (1) or Equation (2)) that was best supported by the data.

| Species | N | 2.5% | 50% | 97.5% | Model |
|---|---|---|---|---|---|
| *Anacamptis morio* | 433 | **−0.1378** | **−0.0715** | **−0.007** | 2 |
| *Anacamptis pyramidalis* | 100 | −0.009 | 0.1233 | 0.242 | 2 |
| *Cephalanthera damasonium* | 137 | −0.1002 | −0.0102 | 0.0662 | 2 |
| *Cephalanthera longifolia* | 182 | −0.066 | −0.0265 | 0.0094 | 2 |
| *Cephalanthera rubra* | 60 | −0.2268 | −0.017 | 0.1976 | 2 |
| *Corallorhiza trifida* | 189 | **−0.4227** | **−0.2633** | **−0.1144** | 2 |
| *Cypripedium calceolus* | 83 | −0.1162 | −0.0157 | 0.0746 | 2 |
| *Dactylorhiza incarnata* subsp. *incarnata* | 641 | −0.2437 | −0.1155 | 0.0086 | 2 |
| *Dactylorhiza maculata* subsp. *fuchsii* | 182 | **−0.0851** | **−0.0491** | **−0.0176** | 2 |
| *Dactylorhiza maculata* subsp. *maculata* | 391 | −0.1085 | 0.0153 | 0.1329 | 2 |
| *Dactylorhiza majalis* subsp. *majalis* | 928 | −0.0439 | 0.012 | 0.0666 | 2 |
| *Dactylorhiza majalis* subsp. *integrata* | 33 | −0.1851 | 0.3126 | 0.8105 | 2 |
| *Dactylorhiza majalis* subsp. *purpurella* | 158 | −0.1351 | −0.0363 | 0.058 | 2 |
| *Dactylorhiza sambucina* | 231 | **−0.2581** | **−0.1358** | **−0.0245** | 2 |
| *Epipactis atrorubens* | 38 | **−0.0823** | **−0.0599** | **−0.038** | 1 |
| *Epipactis helleborine* subsp. *helleborine* | 652 | **−0.1575** | **−0.1173** | **−0.0781** | 2 |
| *Epipactis helleborine* subsp. *neerlandica* | 20 | **0.0058** | **0.0228** | **0.0404** | 1 |
| *Epipactis leptochila* | 217 | **−0.1834** | **−0.1208** | **−0.0608** | 2 |
| *Epipactis palustris* | 310 | −0.3607 | −0.1382 | 0.0771 | 2 |
| *Epipactis phyllanthes* | 232 | **−0.2395** | **−0.1625** | **−0.0967** | 2 |
| *Epipactis purpurata* | 247 | −0.0742 | −0.0352 | 0.0014 | 2 |
| *Epipogium aphyllum* | 26 | −0.1018 | 0.0223 | 0.1545 | 2 |
| *Goodyera repens* | 45 | −1.3174 | −0.5082 | 0.2319 | 2 |
| *Gymnadenia conopsea* subsp. *conopsea* | 63 | **−0.2497** | **−0.1098** | **−0.0264** | 2 |
| *Gymnadenia conopsea* subsp. *densiflora* | 20 | −0.099 | 0.0926 | 0.287 | 2 |
| *Hammarbya paludosa* | 122 | **−0.4299** | **−0.2721** | **−0.1274** | 2 |
| *Herminium monorchis* | 131 | **−0.3406** | **−0.1741** | **−0.004** | 2 |
| *Liparis loeselii* | 408 | −0.124 | −0.0418 | 0.0391 | 2 |
| *Neotinea ustulata* | 32 | −0.1095 | 0.0433 | 0.1952 | 2 |
| *Neottia cordata* | 31 | −0.6676 | −0.1139 | 0.4011 | 2 |
| *Neottia nidus-avis* | 163 | **−0.116** | **−0.0576** | **−0.002** | 2 |
| *Neottia ovata* | 447 | **−0.08** | **−0.0411** | **−0.0054** | 2 |
| *Ophrys apifera* | 21 | −0.1546 | 0.7153 | 1.5831 | 2 |
| *Ophrys insectifera* | 18 | **0** | **0.0344** | **0.0689** | 2 |
| *Orchis mascula* | 795 | **−0.0678** | **−0.0361** | **−0.005** | 2 |
| *Orchis militaris* | 6 | **−1.9705** | **−1.085** | **−0.4714** | 2 |
| *Orchis purpurea* | 140 | −0.0725 | −0.0272 | 0.0208 | 2 |
| *Platanthera bifolia* subsp. *bifolia* | 281 | **−0.3781** | **−0.2153** | **−0.074** | 2 |
| *Platanthera bifolia* subsp. *latiflora* | 82 | **−0.1708** | **−0.1162** | **−0.0683** | 2 |
| *Platanthera chlorantha* | 315 | −0.1442 | 0.059 | 0.2744 | 2 |
| *Pseudorchis albida* | 181 | **−0.0725** | **−0.0606** | **−0.049** | 1 |

Of the 41 orchids, 18 showed a significant decrease in abundance, while 2, *Epipactis helleborine* subsp. *neerlandica* and *Ophrys insectifera*, showed a significant increase in abundance. The abundance of the remaining 21 orchids did not show a significant change (Table 3). In order to ensure that the reported trends are general in both time and space and not due to extraordinary years or sites with uncharacteristic management practices, we added the supplementary constraint that at least 50 observations were needed for making robust and general inferences on the change in orchid abundance. Under this added constraint, 16 orchids showed a significant decrease, while none were found to be increasing (Table 3). In Figure 2, the population decrease for four of the monitored orchids is illustrated.

The effects of the estimated pressures on the change in abundance are shown in Table 4. Generally, there was a significant negative effect of overgrowing with tall-growing herbs and shrubs on the abundance of orchids. The effects of the remaining three pressures were not statistically significant.

**Table 4.** The percentiles of the marginal distribution of the estimated posterior distribution of the effect of the mean estimated pressures at each site on the site-specific random time coefficients, $\delta_i$, of model 2.

| Effect | 2.5% | 50% | 97.5% |
|:---:|:---:|:---:|:---:|
| Intercept | −0.061 | 0.023 | 0.108 |
| Intensity of livestock grazing | −0.009 | 0.016 | 0.040 |
| Intensity of forest management | −0.031 | 0.006 | 0.043 |
| Overgrowth with tall-growing herbs and shrubs | −0.052 | −0.029 | −0.006 |
| Public disturbance | −0.038 | −0.004 | 0.030 |

In Figure 3, the results of three of the more noticeable changes in orchid abundance are presented. The orchids *Anacamptis morio*, *Dactylorhiza sambucina*, and *Herminium monorchis* all show a significant decreasing abundance, which may be explained by a typical land use change from extensive livestock grazing, especially with cattle and mowing, towards more intensive farming practices with the use of fertilizers. Notice also that the chosen linear trend model does not always provide an adequate description of the observed trends, e.g., in the case of *Herminium monorchis*, which for unknown reasons showed large inter-annual variation in abundance.

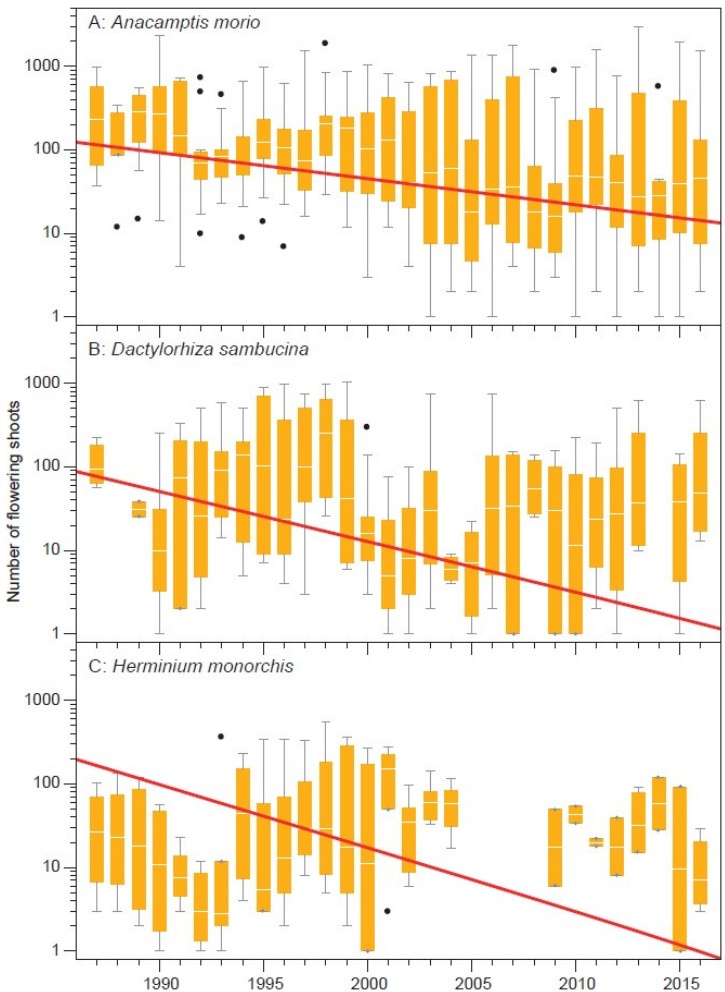

**Figure 3.** The observed number of flowering shoot changes for three orchid species shown as box plots of the mean site abundance (white line: medians, yellow box: 25–75%, whiskers: 2.5–97.5%, points: outliers), where the red line illustrates the estimated annual change in the linear model (2). The shown box plots are a summary of the hierarchical repeated-measure abundance data and cannot be eyeball-fitted to the back-transformed median slope. (**A**) *Anacamptis morio*, (**B**) *Dactylorhiza sambucina*, (**C**) *Herminium monorchis*.

## 4. Discussion

Twenty Danish orchids have shown a significant decrease in abundance over the past 30 years (16, if only orchids with at least 50 observations are considered), thus corroborating the previous observations of declining orchid abundances at European scale [1–3]. Of the declining 16 orchids, 12 are nationally red-listed [4], status in parentheses): *Anacamptis morio* (NT), *Corallorhiza trifida* (VU), *Dactylorhiza sambucina* (EN), *Epipactis atrorubens* (NT), *E. leptochila* (NT), *E. palustris* (NT), *Gymnadenia conopsea* subsp. *conopsea* (CR), *Hammarbya paludosa* (EN), *Orchis purpurea* (NT), *Platanthera bifolia* subsp. *bifolia* (NT), *P. bifolia* subsp. *latiflora* (EN), *Platanthera chlorantha* (NT), and *Pseudorchis albida* (CR). The four not red-listed, declining orchids are *Epipactis helleborine* subsp. *helleborine*, *Neottia nidus-avis*, *N. ovata*, and *Orchis mascula* (Table 1).

There is no clear pattern between the significant decline of the 16 orchids and their abundance, biology, and habitat. Both some of the very rare and of the more common taxa in Denmark are represented among the 16 orchids with both tuberous and rhizomatous subterranean parts as well as *Hammabya paludosa* with pseudobulbs. Of the 16 orchids, 7 are confined to light-open habitats, e.g., grassland, dune slacks, rich or poor fens, and blanket bogs. The remaining orchids grow on both

light-open and shaded habitats. Thus, our results do not corroborate the findings of Vogt-Schilb et al. [45], where the abundance of shade-intolerant orchids declined more sharply than that of shade-requiring orchids on the Mediterranean island Corsica.

Seven threatened orchids did not show a significant decrease in abundance with the red-list category in parenthesis: *Cephalanthera damasonium* (VU), *C. longifolia* (EN), *C. rubra* (CR), *Cypripedium calceolus* (VU), *Herminium monorchis* (EN), and *Liparis loeselii* (EN). These are all species that are either red-listed because of a very limited, national distribution or because their main habitats are in decline, which is also a significant factor considered in the red-listing process. Furthermore, the result of our analysis shows a significant decrease in abundance among seven orchids categorized as least concern (LC): *Dactylorhiza incarnata* subsp. *incarnata, D. maculata* subsp. *fuchsii, Epipactis phyllanthes, E. purpurata, Neottia nidus-avis, N. ovata,* and *Orchis mascula.* This incongruence can probably be explained by the fact that they are all still relatively widespread in Denmark [44]. However more importantly, in our study only a subset of the monitored Danish populations were included compared to the number of estimated sites of Danish orchids in Table 2. Hence, most populations of these species are probably not in decline nationally, but only locally, and therefore our results regarding these species should be interpreted with this in mind.

Two of the 45 monitored Danish orchids, *Epipactis helleborine* subsp. *neerlandica* and *Ophrys insectifera* have shown a significant increase in abundance. The likely reason for the increase of *E. h.* subsp. *neerlandica*, which is not common and nationally red-listed in least concern (LC), is that the orchid grows in calciferous dune slacks in the Danish coastal zone that are not heavily affected by the various pressures that other orchids are confronted with. Moreover, the coastal zone and the dune areas are legally protected to prevent human changes of the coastal habitats. In Denmark, there is only one population of *O. insectifera* growing in a light-open forest on calciferous soil on Zealand, nationally red-listed as critical endangered (CR). The forest of 65 ha is now owned by "Danmarks Naturfond", a private, independent foundation under "Danmarks Naturfredningsforening". The aim of the foundation is to preserve landscapes and cultural historical values, to protect plant and animal life, and to provide recreational areas for the population. Volunteers perform annual nature management to keep the sites in the forest for the *O. insctifera* light-open.

According to our analyses, the observed general decline in orchid abundance may be partially caused by overgrowth with tall-growing herbs and shrubs, which most likely is caused by changes in the traditional farmland-use, especially livestock grazing or mowing. The speed of the natural vegetation dynamics may in many instances be increased by the intensive agricultural use of fertilizers and manure on neighboring fields to orchid sites. This inference is corroborated by the results from Timmermann et al. [54], who found that reduced livestock grazing most likely was an important factor in explaining the decline in abundance of some plant species. On the contrary, we did not find a significant effect of public disturbance and forest management on the observed changes in orchid abundance in this study.

Many orchids have relatively complex life-cycles and quite narrow ecological requirements for their habitat. A number of orchid species are pollinated by one or two insect species [55], but most are pollinated by a more diverse insect fauna. For instance, *Ophrys insectifera* are pollinated by males of the digger wasps *Argogorytes mystaceus* and *Argogorytes frageii*, while the flowers of *Cypripedium calceolus* are pollinated by females of the genus *Andrena* and *Halictidae* [5]. Hence, a decline in such pollinating insects could also explain the decline in their dependent orchid species. In addition, most orchid species depend on certain mycorrhizae to germinate and thrive, and are therefore prone to reduction in the availability of these fungal relationships. For example, a recent study showed that the landscape-scale distribution of four European orchid species, *Cephalanthera rubra, Epipactis atrorubens, E. helleborine,* and *Neottia nidus-avis*, are primarily restricted by availability of fungal associates [7].

## 5. Conservation and Protection of Danish Orchids

There are several legal and administrative tools that can be applied to protect the population of the Danish orchids and their habitats. (1) All orchids are protected against harming and picking the plants, collection of their seed, and digging by the Danish species conservation act no. 1466 of 6th December 2018. (2) The legal protection of specific orchid sites. (3) Paragraph three in the Danish nature protection law that states a general protection of the light-open, terrestrial nature types, dry grassland, heathland, fens, and fresh and saline meadows over a certain size against changes of their present state. (4) The establishment of National Parks. (5) The implementation of the Habitats Directive (the European Council Directive 92/43/EEC of 21 May 1992) on the conservation of natural habitats and of wild fauna and flora.

However, none of the tools are sufficient to protect Danish orchids. (1) Despite general legal protection with a criminal frame, Danish orchids are continuously picked or dug up where even a tight fence cannot prevent the theft of flowering or entire clones of *Cypripedium calceoulus*, and although the thefts are reported to the police, no one has been charged. Presumably, the thefts of orchids normally take place during times of the day where the thieves can fulfill their illegal mission undisturbed. On the other hand, information from the Danish authorities on the protected species is scant and often random. (2) A few sites have been legally protected because of the presence of orchids. An example is the limestone cliffs and forest on the island of Møn, which have been protected since 1921, among other things because of the great amounts of different orchids found here. Information stands inform the public of the protected species at the entrances to the protected areas of Møn. (3) The protection of light-open Danish nature types is a general tool. If a landowner wishes to alter the state or use of an area with a protected nature type, s/he must seek permission from the local community. Unfortunately, the general protection cannot prevent farmland-use of a legally protected area from being stopped, allowing the natural vegetation dynamics to continue. This may result in overgrowth with tall herbs and shrubs and outcompeting of the low-growing species, such as orchids. (4) In Denmark, five national parks have been designated. Of these, none have been appointed in order to specially protect orchids. Voluntarily, the staff of National Park Mols Bjerge in Jutland has chosen to map the distribution of 20 key vascular plant species in order to map their distribution and to gain further information for their protection. *Orchis mascula* is one of the target species. The project is still running. (5) Apart from habitat type 6210 "semi-natural dry grasslands and scrubland facies on calcareous substrates (*Festuco-Brometalia*)", where "important orchid sites" are prioritized according to the Habitat Directive, no habitat type has been appointed because of the presence of orchids. In addition, two Danish orchids, *Cypripedium calceolus* and *Liparis loeselii*, are listed in the Habitats Directive Annex II and IV. This implies for species listed on Annex II that the core areas of their habitat are designated as sites of community importance and included in the Natura 2000 network. These sites must be managed in accordance with the ecological needs of the species. According to the species listed in Annex IV, a strict protection regime must be applied across their entire natural range within the EU, both within and outside Natura 2000 sites.

As of yet, no specific national conservation strategy for the protection of orchids has been worked out in Denmark. A national conservation strategy should be developed in collaboration between scientists and authorities, i.e., the Danish Nature Agency. To work out an effective national conservation strategy, it is necessary to understand the orchids' biology, which requires further research into areas including pollination, mycorrhizal associations, population genetics, and demographics, as Fay [56] has pointed out.

**Supplementary Materials:** The following are available online at http://www.mdpi.com/1424-2818/12/6/244/s1, Appendix SA: Citizen Science in Denmark, Appendix SB: The National Orchid Monitoring Program, Appendix SC: Surveyors, Appendix SD: Statistical model.

**Author Contributions:** Conceptualization, All; methodology, All; formal analysis, C.D.; investigation, All; resources, P.W.; data curation, P.W.; writing—original draft preparation, C.D.; writing—review and editing, All; visualization, All; project administration, P.W.; funding acquisition, P.W. All authors have read and agreed to the published version of the manuscript.

**Funding:** This research was funded by the Danish 15. Juni Fonden.

**Acknowledgments:** We sincerely thank all the surveyors (see Supplementary Materials, Appendix SC) of the National Orchid Monitoring Program for collecting the data used here. The language was improved by Charlotte Kler. Three anonymous referees have provided very useful comments on a previous version of the manuscript.

**Conflicts of Interest:** The authors declare no conflict of interest. The funders had no role in the design of the study; in the collection, analyses, or interpretation of data; in the writing of the manuscript, or in the decision to publish the results.

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
