# Peer review of "Changes in the Abundance of Danish Orchids over the Past 30 Years"

_diversity, doi:10.3390/d12060244_

Round 1
Reviewer 1 Report
Authors of the reviewed manuscript analyse changes of orchids populations existing in Denmark and observed during the 30 years (1987- 2016). Data were collected from 440 sites within the National Orchid Monitoring Program. Presented data are valuable and are good example of monitoring of rare and endangered species at the country level. The data are valuable due to conservation point of view and are useful for planning protection actions. Despite data and analyses are valuable, I have some questions and suggestions to the present version of the manuscript.
- Totally Authors analyzed 51 orchid taxa in 440. It would be interesting to include information about number of sites/populations of a given species and range of populations sizes (max.-min.). Such data should be included in Table 1.
- My second main remark concerns Discussion chapter. It is the weakest part of the manuscript. In my opinion in the present version is poor and partially repeats or completes results. In Discussion authors should deeper and more precisely analyze problem studied, for example more precisely identify causes of negative changes in orchid populations in Denmark. Propositions of conservation action should be included.
Minor remarks:
Row 25. "taxonomic level" - should be rather "taxonomic status", also in the Table 1.
Row 99. "Fifty one terrestrial orchids" - should be "Fifty one terrestrial orchid taxa"
Rows 101-102. Are authors sure that Ophrys apifera and Orchis militaris, recently appeared in Denmark in 2004 and 1981? May be they were present for longer time, and were only found in 2004 and 1981 in Denmark? If they really appeared recently, it is interesting to know their origin. Who found the first localities of these orchids?
Row 140. Is Pla bi flat – P. b. subsp. latiflora, should be Pla bif lat – P. b. subsp. latiflora
Fig. 2. What indicate different colours?
Reviewer 2 Report
The conservation of biodiversity is becoming a crucial issue, and orchids represent a remarkable biodiversity whose monitoring has an exemplary value for other taxonomic groups, particularly floristic groups. Moreover, to my knowledge, this is the first information of this type in Denmark. This study is made possible by an impressive work of participatory science that has made it possible to monitor 440 sites over 30 years, which is rare on a European scale. The use of the Bayesian framework is very appropriate to analyze such kind of survey data and allow to test the effect of different environmental factors to explain the observed variation of presence species per species.
This work does not present any scientific error but rather suffers from an insufficient interpretation of the results presented here. Here are several avenues for improvement:
This work presents first of all a good review on a European scale of the analyses of variations in orchid occurrence in Europe as well as long-term monitoring. For analyses of diachronic variations in orchid occurrence, the results of these two articles should be added:
Jacquemyn, H., Brys, R., Hermy, M., Willems, J.H., 2005. Does nectar reward affect rarity and extinction probabilities of orchid species? An assessment using historical records from Belgium and the Netherlands. Biological Conservation, 121, 257–263.
Vogt-Schilb H., Pradel R., Geniez P., Hugot L., Delage A., Richard F., Schatz B. 2016. Effects of habitat change, population size and distribution patterns on orchid dynamics: a study based on a 27-year interval in Corsica. Annals of Botany 118 (1): 115-123.
The introduction could begin more broadly by presenting the general decline in biodiversity as a result of global changes. Many human activities have positive or neutral but unfortunately more often negative impacts on biodiversity. The identification of activities with negative impacts is important for reorienting the national conservation strategy. In addition, citizen science allows the monitoring of biodiversity on a large scale and to document changes in the presence of species. In this context, orchids constitute a remarkable biodiversity that can serve as an example for other taxonomic groups.
Globally, I find that the authors do not value sufficiently the interest due to the multi-year monitoring of their network of orchid stations. The studies of orchid occurrence variations (presented in the introduction) allow us to contrast past and present situations. However, multi-year monitoring of orchids provides richer information on the extent of annual variation. Is it possible, for example, to determine whether certain years are harmful to abundance and other years are positive for all orchids?
The end of the introduction deserves to detail more explicitly the different objectives of this study. That is an important point for the clarity of this paper.
L237-238: How explain the case of Herminium monorchis for which the interannual variation of the flowering shoots suggest a relative stability in spite of great variation, while the estimated annual change in the linear model showed a great decrease.
In the results, the authors could have made better use of their database by determining the levels of interannual variation for each species (those with sufficient repetition). See for this point the article Vogt-Schilb et al 2013 and the cited articles therein.
Vogt-Schilb H., Geniez P., Pradel R., Richard F. & Schatz B. 2013. Inter-annual variability in flowering of orchids: lessons learned from 8 years of monitoring in a Mediterranean region of France. European Journal of Environmental Sciences 3: 129-137.
The conclusion is clearly too short, and the authors do not sufficiently exploit the diversity of their results. Here are a number of areas for improvement:
Observed differences IUCN status and the significant decrease or increase (L250-261) are puzzling and they merit to better explained, more especially because Peter Wind is a common author of these works done in a very similar time. These differences also concern almost one third (16 out of 51) orchid taxa present in this country Are these differences explained by variation between expert perception of the orchid abundance for IUCN analysis and the results of the model?
The decline of orchids in Denmark is thus mainly explained by overgrowth with tall-growing herbs and shrubs, and then by a reduction of grazing.
This is interesting because it indicates that this is not irreversible, and that an agricultural policy favoring grazing would improve the conservation of orchids in this country. In the other cases of decline observed in the different European countries, the main cause was rather the increase in forest cover. How can this difference be explained?
L272-273: The authors may add the reference Nieto et al 2014 about the European red list of bees.
Nieto, A., Roberts, S.P., Kemp, J., Rasmont, P., Kuhlmann, M., García Criado, M., et al., 2014. European red list of bees. Luxembourg: Publication Office of the European Union, 98p.
Personally, I know little about the orchid conservation strategy in Denmark, and I imagine that I am far from being the only one. Apart from the red list of orchids, is there a list of protected orchids in Greece? or protected areas with attention to orchidoflora? Is this increase in knowledge of orchids not an opportunity to establish a list of protected species, at least for orchids?
Do protected areas (national parks, Natura 2000 site, ...) allow better protection of orchids? Are there more positive situations with less decline in these areas? Two other exploitations of the environment are also interesting to comment. The fact that forest management and public disturbance explain relatively little of the variation in orchid presence (Tab. 3) shows that they are relatively well managed. It would be interesting for the reader to know what other environmental factors were tested here. What is the effect of urbanization, intensive agriculture, drainage of wetlands, etc.?
n addition, several species situations are interesting to better comment on:
The case of Epipactis helleborine is particularly surprising because subspecies helleborine is in decline while subspecies neerlandica is increasing. Why is this? A similar situation is observed to a lesser degree for the two subspecies of Dactylorhiza maculata (Table 2).
How can we explain that Ophrys insectifera while it is a species with sexual deception), therefore with a rather strong ecological requirement L269-273)? Why is this species in rather open environments increasing while other more forested species are in decline?
How can it be explained that Orchis militaris appeared in 1982 in Denmark and is today the species in greatest decline. Is it the stochastic effect on a small number of populations?
The legend of the figure 2 needs to explain the meaning of the three colors (green, red, black) used in the graph.
L275: Replace Cephalathera by Cephalanthera
Bibliography: Write in italics all latin names cited in the title of references.
Author Response
See attachment to reviewer 1
Reviewer 3 Report
The paper analyses long-term orchid monitoring data. I'm really pleased to read such a nice overview of long-term monitoring where the data are so well analysed. There are only some small things I would like to be corrected/improved.
row 188 you sat flower numbers? probably flowering shoot number
in fig 2 caption I miss meaning of colours. In the text it is but should be also in the caption.
Author Response
See attachment to reviewer 1
Round 2
Reviewer 1 Report
The second version of the manuscript was corrected according to suggestions of reviewers. Despite this, the present version is still unsuitable for publication. First of all, although authors improved Discussion, this chapter is still unsatisfied, in my opinion. It is not coherent and sketchy. It is mixture of general and detailed info. It is not a good idea to start discussion with examples. Similarly, the rest of this chapter is also presentation of a particular cases. Moreover, in the cases of particular species different types of info are given. There are info of different range. For example, authors wrote about abundance of pollinators of C. calceolus and Ophrys insectifera as factor shaping populations size, rows 396-401. What about other species? This problem concerns almost all orchids.
Rows 167-170. Authors wrote The rhizome can divide so that the aerial shoots represent a clone )rather are part of a clone. The rhizome can divide vegetative into several separate units that can form aerial shoots that without digging the rhizome complex up are impossible to separate rather to distinguish from each other and thus functions as a clone.
It is not well write sentences. They don't give precise info. It would be better to say about fragmentation. Shoots arrived from vegetative fragmentation belong to the clone, but function as independent units after fragmentation.
Writing about pollination type Authors should obviously include info from Nilsson papers.
The language of new fragments should be corrected.
Author Response
see uploaded document
